

# BACPHLIP: predicting bacteriophage lifestyle from conserved protein domains

Adam J. Hockenberry and  Claus O. Wilke

Department of Integrative Biology, The University of Texas, Austin, TX, United States of America

## ABSTRACT

Bacteriophages are broadly classified into two distinct lifestyles: temperate and virulent. Temperate phages are capable of a latent phase of infection within a host cell (lysogenic cycle), whereas virulent phages directly replicate and lyse host cells upon infection (lytic cycle). Accurate lifestyle identification is critical for determining the role of individual phage species within ecosystems and their effect on host evolution. Here, we present BACPHLIP, a BACterioPHage LIfestyle Predictor. BACPHLIP detects the presence of a set of conserved protein domains within an input genome and uses this data to predict lifestyle via a Random Forest classifier that was trained on a dataset of 634 phage genomes. On an independent test set of 423 phages, BACPHLIP has an accuracy of 98% greatly exceeding that of the previously existing tools (79%). BACPHLIP is freely available on GitHub (https://github.com/adamhockenberry/bacphlip) and the code used to build and test the classifier is provided in a separate repository (https://github.com/adamhockenberry/bacphlip-model-dev) for users wishing to interrogate and re-train the underlying classification model.

# INTRODUCTION

Bacteriophages play important ecological roles (*Brum et al., 2015*; *Paez-Espino et al., 2016*; *Nishimura et al., 2017*; *Emerson et al., 2018*; *Daly et al., 2019*) and influence both the physiology and evolution of host species (*Touchon, Bernheim & Rocha, 2016*; *Bondy-Denomy et al., 2016*; *Carey et al., 2019*). Additionally, phages possess a number of unique traits (*Bobay & Ochman, 2018*; *Ofir & Sorek, 2018*) that are relevant to ongoing biotechnological and medical applications (*Roach et al., 2017*; *Dedrick et al., 2019*; *Rodriguez-Gonzalez et al., 2020*). The availability of phage genome sequences has expanded greatly in recent years—owing largely to advances in metagenomic technologies  (*Deng et al., 2014*; *Pope et al., 2015*; *Roux et al., 2015*; *Simmonds et al., 2017*; *Mizuno et al., 2019*; *Tisza et al., 2020*).

A particularly important phage phenotype is whether the phage has a temperate or virulent lifestyle (*Bobay, Rocha & Touchon, 2013*; *Harrison & Brockhurst, 2017*; *Li et al., 2020*). Temperate phages are capable of lying dormant within the host cell (lysogenic cycle), whereas virulent phages directly replicate and lyse host cells upon infection (lytic cycle). Phages are extraordinarily diverse and this dichotomy is an over-simplification (*Abedon, 2008*; *Dion, Oechslin & Moineau, 2020*). For instance, phages may chronically infect cells

Corresponding author
Adam J. Hockenberry,
adam.hockenberry@utexas.edu

over long periods of time *without* explicitly integrating into the genome—a process known as pseudo-lysogeny (*Ripp & Miller, 1997*; *Latino et al., 2016*)—whereas the lysogenic cycle itself has tremendous diversity in both mechanisms and outcomes (*Howard-Varona et al., 2017*). Additionally, some mobile genetic elements blur the lines between temperate phages and plasmids, subverting our basic classification systems (*Pfeifer et al., 2021*). Nevertheless lifestyle classification into temperate and virulent categories is broadly recognized and important in numerous contexts (*Mavrich & Hatfull, 2017*; *Moura de Sousa et al., 2021*).

Phage lifestyle should ultimately be determined experimentally, but such labor-intensive approaches are impractical for the entirety of newly discovered phage genomes (*Camarillo-Guerrero et al., 2021*). Using computational methods, *McNair, Bailey & Edwards (2012)* developed a Random Forest classifier (PHACTS) to predict lifestyle based off of sequence similarity to a set of query proteins (randomly selected from the phage proteome training set). While an important advance, the PHACTS model was trained and tested on a set of phages that were available in 2012—a total of 227. More recently, *Mavrich & Hatfull (2017)* compiled a dataset of >1,000 phage genomes with lifestyle annotations and describe a distinct classification method based on detecting the presence of specific protein domains. Their approach reported a high prediction accuracy of ~96%, but training and testing sets were not explicitly defined. Moreover, software development was not the focus of that study, and their lifestyle classification tool is not available for general use.

Here, we combined the distinct approaches used in previous studies (*McNair, Bailey & Edwards, 2012*; *Mavrich & Hatfull, 2017*) to create an up-to-date, user-friendly software package for predicting phage lifestyles based on genome sequence input data. BACPHLIP (BACterioPHage LIfestyle Predictor) is a python library with an optional command-line interface that relies on the HMMER3 software suite (*Eddy, 2011*) to identify the presence of a set of lysogeny-associated protein domains. BACPHLIP initially assumes that the input genome (nucleotide) sequence is from a fully complete, virulent phage and the presence and pattern of specific protein domains can override this assumption and result in a temperate classification. BACPHLIP is fully open-source and achieves high prediction accuracy (98.3%) on an independent test set of 423 phages—greatly exceeding the performance of prior state-of-the-art methods.

## MATERIALS AND METHODS

### Selection of lysogeny-associated protein domains

We developed BACPHLIP by first searching the Conserved Domain Database (*Lu et al., 2020*) (accessed: 03/2020) for protein domains that we hypothesized to be enriched in temperate phages (i.e. mechanistically involved in lysogeny). We searched the 'description' field using the following search terms: 'integrase', 'excisionase', 'recombinase', 'transposase', 'lysogen', and 'temperate'. We additionally included a case sensitive search for 'parA |ParA |parB |ParB' due to its short length and potential overlap with many common words. These genes encode partitioning proteins that are frequently found on plasmids and in select temperate phage genomes, and were included in our search strategy due to their use in *Mavrich & Hatfull (2017)*, which brought their importance to our attention. We did not
include a broad set of protein domains in our search strategy to ensure interpretability of our results and to limit the possibility of over-fitting.

Collectively, we identified 371 protein domains that formed the starting set of putatively useful protein domains. We stress that the 'description' field of the selected domains contained one or more of the above words at some point within it, but these domains may or may not *actually* be proteins or enzymes with any of the hypothesized functions. Some fraction of the initial set of 371 domains were thus likely to be unhelpful for the task of delineating temperate from virulent phages, due either to erroneous annotations or mis-classification via our simple keyword-based search strategy.

After removal of protein domains that were present in two or fewer *training* set phage genomes (see below for training set definitions) or which were actually *more* prevalent in the annotated virulent phage genomes (again, only considering phage genomes from the training dataset), we established a condensed dataset of 206 putatively useful lysogeny-associated protein domains. At this stage, we still did not know if any/all of these 206 domains would be useful for delineating temperate and virulent phages, which is why we next used this data as input into a Random Forest classifier that we hypothesized would disregard unimportant features (domains) and detect higher-level patterns in the data.

## Fitting a random forest classifier to training set data

To train and test a lifestyle classification model, we leveraged 1,057 phages with empirically established lifestyles that were collected by *Mavrich & Hatfull (2017)*. While some of these phage lifestyles may be mis-annotated based on experimental errors or inconsistencies in reporting, our study assumes that these phages form a set of ground-truth data in which to build upon. We took this initial dataset and randomly split it into two separate groups (using a 60:40 split, 634 and 423 phages) that we refer to as training and testing sets. At this stage, the testing set was fully set aside for the remainder of any discussion of model training/development.

For each genome sequence in the training set, we created a list of all possible 6-frame translation products $\geq$ 40 amino acids. Next, we used HMMER3 (*Eddy, 2011*) to search for the presence of various protein domains discussed above, resulting in a vector for each phage describing the presence (1) or absence (0) of each domain. We used this information to filter the initial set of 371 putatively useful protein domains down to 206 (as described above).

We next fit a Random Forest classifier to the labeled training set of phage genomes using cross-validation to tune model hyper-parameters. During hyper-parameter tuning phase, the training set (634 genomes) was randomly split into individual training and validation sets. The accuracy of various hyper-parameters was tested on the validation sets (after being fit to the training set) and for each set of hyper-parameters we randomly selected 20 different training/validation sets drawn from the same set of 634 training set genomes. Specifically, within the `scikit-learn` Random Forest framework, we evaluated 'bootstrap' (True, False), 'class_weight' (balanced, balanced_subsample), 'min_samples_leaf' (1, 2), 'n_estimators' (range(10, 105, 5)), and 'max_depth' (range(10, 42, 2)). We used 'GridSearchCV' to evaluate all possible combinations of these parameters (using 'f1'as

the scoring function). We note briefly that the 'class_weight' parameter in particular was used to assign variable weights given that the number of temperate and virulent phages in our training set were uneven. The 'balanced_subsample' parameter should correct for this un-evenness and ensure the accuracy of a model despite the (slightly) imbalanced classes.

At the end of the hyper-parameter search phase, we were left with 20 different measurements of validation set accuracy (where each validation set was independently drawn from the larger training set) for each possible hyper-parameter combination. We selected the 'best' model by choosing the hyper-parameters that yielded the *highest minimum accuracy* across the 20 independent validation set tests. The parameters of that model were then re-fit to the *entire* training set of data to become BACPHLIP. The final BACPHLIP model selected: 'bootstrap = False', 'class_weight = balanced_subsample', 'min_samples_leaf = 1', 'n_estimators = 80', and 'max_depth = 40'. All other parameters followed `scikit-learn` defaults as of version 0.23.1.

Critically, at this stage none of the testing set data (423 phages) were used for any purpose, which makes it a fully independent dataset to test how well the BACPHLIP model performs. Indeed, the held out 423 phage genomes could have come from the same initial starting dataset or somewhere else entirely and this distinction should make no difference to our ability to assess the accuracy of BACPHLIP. We therefore report (and emphasize) most of our results in terms of this independent testing set of phages.

While this testing set of phage genomes was randomly chosen for the purposes of developing/evaluating BACPHLIP, we tested all of the different methods (including PHACTS and the method developed by *Mavrich & Hatfull (2017)*) on the same testing set of genomes. However, we note that it is possible (likely) that some genomes in what we define as our *testing* set may have been a part of the *training* set for these other methods. Our approach in evaluating the accuracy of other methods and comparing them to BACPHLIP is thus highly conservative since any phage genome that was previously used to train the previous approaches should ideally be excluded when evaluating the effectiveness of these models.

## Constructing a phylogenetically-independent testing set of phage genomes

Our testing set of 423 genomes was not used for tuning the BACPHLIP model in any way, but an important issue to consider is that biological data is often not fully independent (*Bobay & Ochman, 2018*). Thus, it is entirely possible that some fraction of our 423 testing set genomes have very close relatives in the training set of data. To ascertain the effect of phylogenetic structure in our dataset—which could inflate testing set accuracy metrics—we clustered all labeled phages using FastANI and the CD-HIT algorithm (*Fu et al., 2012*; *Jain et al., 2018*), selected single representatives from each cluster, and evaluated accuracy separately on testing set data for which there were no genomes in the training set with >80% sequence identity across >80% of the genome. This procedure resulted in a set of 157 of stringently defined, independent testing set genomes that we could use to more conservatively evaluate expected the performance of BACPHLIP on novel phage genomes.

## Assessing existing models

For assessing PHACTS, we report the lifestyle as the category with the highest probability *regardless* of the confidence, reasoning that "no prediction" should be viewed as an error. BACPHLIP similarly reports a probability that the phage is either temperate or virulent, and we also report the lifestyle with the highest probability regardless of its actual value.

For the *Mavrich & Hatfull (2017)* method, we rely on the pre-computed model predictions provided by the authors alongside their set of empirically determined annotations that they separately collected. In order to allow out-of-sample prediction and to study a larger dataset of phages, *Mavrich & Hatfull (2017)* built a classifier and the predictions of this classifier were provided for all phages (including those in the empirically determined dataset).

## Data and code availability

The BACPHLIP tool is freely available on GitHub (https://github.com/adamhockenberry/bacphlip) and consists of the trained random forest model, alongside supporting functions that are required to integrate the full-pipeline from genome inputs ('.fasta' formatted) to prediction outputs. The code used to build and test the classifier is provided in a separate repository (https://github.com/adamhockenberry/bacphlip-model-dev) for users wishing to interrogate and re-train the underlying classification model and this repository also contains processed data for replicating manuscript figures. Finally, the full underlying dataset has been permanently archived with zenodo (https://doi.org/10.5281/zenodo.4058664).

## RESULTS

### Evaluating the accuracy of lifestyle prediction classifiers

BACPHLIP is a Random Forest classifier that uses the presence/absence of lysogeny-associated protein domains to predict whether a given phage genome input is either temperate or virulent (see Materials and Methods for full model details). The BACPHLIP model was trained on a set of previously labeled phage genomes compiled by *Mavrich & Hatfull (2017)*, and we tested the performance of this approach on an independent test set of 423 phages (240 temperate and 183 virulent) that were withheld for the entirety of model training and development (also compiled by *Mavrich & Hatfull (2017)*). On this independent testing set, BACPHLIP achieved a low error rate of only 1.7% (equivalent to a 98.3% classification accuracy with 415/423 correct predictions, Fig. 1). On the same testing set of phages, the accuracy of BACPHLIP exceeded that of the previously existing PHACTS software (*McNair, Bailey & Edwards, 2012*), as well as the results reported by *Mavrich & Hatfull (2017)*: 21% and 4.5% error rates, respectively (equivalent to classification accuracies of 79% and 95.5%, Fig. 1).

To ascertain the effect of phylogenetic structure in our dataset (which could inflate accuracy metrics), we clustered all labeled phages and evaluated accuracy separately on testing set data for which there were no genomes in the training set with >80% sequence identity across >80% of the genome (see Materials and Methods for details). Although the training set contained only distant phylogenetic relatives to these 157 genomes, BACPHLIP
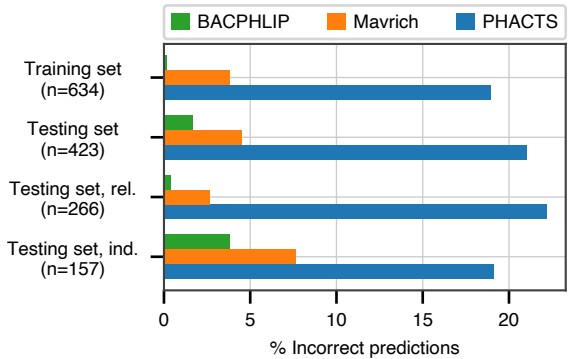

**Figure 1** **Classification accuracy of each compared method across all data sets analyzed.** The labels 'rel.' (related) and 'ind.'(independent) refer to subsets of the testing set with (and without) related genomes contained in the training set.

still achieved 96.5% accuracy (82% and 93% for PHACTS *McNair, Bailey & Edwards (2012)* and *Mavrich & Hatfull (2017)*).

## Alternative measurements of model performance

While the previous results highlight the superior performance of BACPHLIP compared to existing approaches, both accuracy and error rate can be misleading in classification tasks—particularly when the dataset being evaluated is highly imbalanced towards one category. Given that our classes are relatively well balanced, accuracy is an appropriate and easy to interpret measurement. However, a number of other metrics have been developed to deal with the nuance of binary classification problems. In Table 1 we show our accuracy results alongside three other measures of model performance (balanced accuracy, Matthew's Correlation Coefficient, and F1-score) for all three phage classification methods on the testing set data. While the numbers capture subtly different effects, the end result is always the same: (i) BACPHLIP consistently outperforms all other methods and (ii) on an absolute scale, its performance is indicative of a highly accurate and robust model.

## Exploration of classification errors

Each of the above described metrics summarize classifier performance in a single number, but different types of classification errors can occur and the nature of these errors may provide insight that can aid in future model improvements. BACPHLIP makes only 7 incorrect predictions on the testing set data, but it is useful to know whether these incorrect predictions are always in one direction (i.e., classifying a true temperate phage as virulent or classifying a true virulent phage as temperate). The gold standard for depicting binary classification results is a confusion matrix, where the matrix rows indicate the true phage lifestyles and the matrix columns are predictions. Table 2 shows these results for BACPHLIP, highlighting that 6 of the 7 errors occur when BACPHLIP classifies true temperate phages as virulent. These results are quite similar for Mavrich (Table 3), which makes more than twice as many errors but does so largely in the same general pattern as BACPHLIP. We note that this similarity is perhaps not surprising given that both of these

**Table 1  Performance measures for the testing set data (*n* = 423).** For all measures, higher values represent better model predictions. Balanced accuracy considers the uneven balance of classes ('temperate' and 'virulent') and was calculated with 'adjusted = True' such that random guessing would have a score of 0 and a perfect model would have a score of 1. 'MCC' stands for Matthew's correlation coefficient.

|  | BACPHLIP | Mavrich | PHACTS |
|---|---|---|---|
| Accuracy | 0.983 | 0.955 | 0.79 |
| Balanced accuracy | 0.97 | 0.917 | 0.528 |
| MCC | 0.967 | 0.911 | 0.586 |
| F1-score | 0.985 | 0.939 | 0.837 |

**Table 2  Testing set confusion matrix for BACPHLIP.** Actual classes are depicted in rows, predicted classes in columns.

|  | Predicted class | |
|---|---|---|
|  | Virulent | Temperate |
| Virulent | 182 | 1 |
| Temperate | 6 | 234 |

**Table 3  Testing set confusion matrix for *Mavrich & Hatfull (2017)*.** Actual classes are depicted in rows, predicted classes in columns.

|  | Predicted class | |
|---|---|---|
|  | Virulent | Temperate |
| Virulent | 180 | 3 |
| Temperate | 16 | 224 |

methods rely on targeted identification of protein domains associated with lysogeny and that both were trained on the same dataset of phages with known lifestyle annotations—which were compiled by *Mavrich & Hatfull (2017)* separately from the predictive model that they ultimately built for the purpose of out-of-sample prediction.

The incorrect classification of several true temperate phages as virulent (6 instances for BACPHLIP) suggests that either these particular temperate phages have incomplete genomes (and are thus missing critical lysogenic machinery for bioinformatic methods to identify) or more likely that the lysogenic machinery that they do have is novel, uncharacterized, or highly diverged and thus not picked up by the (limited) set of protein domains that we used to build BACPHLIP. A deeper understanding of sequence level diversity within integrase, recombinase, excisionase, *etc.* families may eventually allow us to improve our method and correct these errors.

By contrast, the confusion matrix for PHACTS is heavily biased in the opposite manner (Table 4). PHACTS makes many more errors compared to the other methods, and its errors are heavily biased towards classifying true virulent phages as temperate. The precise reason for this bias is unknown but the information could be valuable for future improvements to the PHACTS methodology.

As one final test of potential factors that impact model accuracy, we investigated whether genome size had any impact on BACPHLIP's testing set accuracy. The results

**Table 4** **Testing set confusion matrix for PHACTS.** Actual classes are depicted in rows, predicted classes in columns.

| | Predicted class | |
| --- | --- | --- |
| | **Virulent** | **Temperate** |
| Virulent | 105 | 78 |
| Temperate | 11 | 229 |

were insignificant (Wilcoxon rank-sum test, $p = 0.45$) when comparing the genome sizes of incorrectly and correctly classified phages in the testing set data. However, given the small number of incorrect test set predictions (7 out of 423) we note that the magnitude of the difference in sizes between incorrectly and correctly predicted phage genomes would have to be very large to be detected. At present, we do not have any reason to suspect that genome-size will play an important role in the ability to classify a given phage, provided that the genome is complete.

## Relative feature importance in the BACPHLIP model

Although we have demonstrated the strong performance of BACPHLIP relative to existing methods, the analyses thus far have not yet shown *why* or *how* BACPHLIP makes individual predictions—a common and difficult problem in understanding machine learning models such as Random Forest classifiers. After fitting the BACPHLIP model to the training data, we found (as with many machine learning tasks) that the overall importance of the 206 features (individual protein domains) was highly variable.

Figure 2A shows that the distribution of feature importance values is highly skewed. Six features have a final feature weight of 0 indicating that they were completely discarded by the classifier model. The 'Top 20' protein domains (ranked in order of feature importance) accounted for 59% of the model weights and the 'Top 50' protein domains accounted for 85% (Fig. 2B). Table 5 depicts the relevant number of domains in each category whose description contains the various search terms that we used at the outset. From this data, it is clear that the most important search terms were 'integrase' and 'recombinase'. However, each term in our search strategy apparently contributed something meaningful as the 'Top 50' most important domains contained at least one domain matching each term.

Many domain descriptions contain several search terms such that it is unclear from this whether each term made a *unique* contribution. We separately looked at the 'Top 50' features, this time filtering out any description that contained matches to multiple terms. Only 'excisionase' had zero relevant domain hits given these constraints, which is not surprising as this was the term with the fewest number of overall domain matches. These results highlight that any conserved domain search strategy that omitted one of these search terms would have invariably produced a worse model. Of course, it is likely that many terms that we *did not* consider may result in further improvements to future versions of BACPHLIP.

## Leveraging prediction confidence to improve accuracy

Rather than outputting a simple prediction (temperate or virulent), BACPHLIP outputs a probability of belonging to either class. This is similar to the PHACTS model (*McNair,*
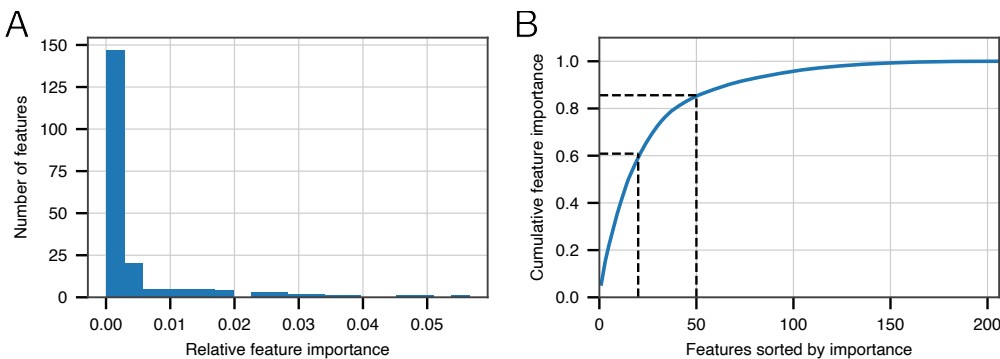

**Figure 2 Feature importance in the BACPHLIP model.** (A) The distribution of importance values for individual features is highly skewed. (B) Cumulative distribution highlighting that the top 20 and top 50 features account for 59% and 85%, respectively, of the overall feature importance values.

**Table 5 For each search term used to identify putatively important conserved protein domains, we show the number of domain descriptions that contain this term for various categories including: (i) the starting set, (ii) the input to the Random Forest model, (iii) the top 50 and (iv) the top 20 features after model fitting.** Note that column entries will not sum up to the *n* depicted at the top of each column as many descriptions contain multiple search terms.

| Search term | Starting ($n = 371$) | Model ($n = 206$) | Top 50 | Top 20 |
|---|---|---|---|---|
| integrase | 101 | 72 | 24 | 13 |
| excisionase | 5 | 4 | 2 | 0 |
| recombinase | 72 | 52 | 28 | 17 |
| transposase | 143 | 70 | 14 | 3 |
| lysogen | 23 | 10 | 2 | 1 |
| temperate | 11 | 10 | 3 | 0 |
| parA |ParA |parB |ParB | 65 | 29 | 7 | 0 |

*Bailey & Edwards, 2012*), which is also based on a Random Forest classifier. In our evaluation, we have selected the lifestyle for an individual phage by considering simply which lifestyle is predicted with >50% probability. However, the confidence of an individual prediction is much stronger for lifestyles that are predicted with 99% probability versus those that are predicted with only 51% probability; depending on the particular problem and use case, individual users may wish to limit their analyses to only high confidence predictions.

Of the 423 testing set genomes, BACPHLIP predicted a lifestyle with ≥95% probability for 333 genomes (78.7%, Figs. 3A, 3C). The recommended methodology for assigning high confidence to PHACTS predictions is slightly different, but we followed the author guidelines and found that PHACTS confidently predicted the lifestyle for 199 of the 423 testing set genomes (47%, Figs. 3B, 3C). Of the 333 high confidence BACPHLIP predictions, there was only one classification error (corresponding to an accuracy of 99.7%). By comparison, BACPHLIP made 6 erroneous predictions in the 90 genomes that were predicted with <95% probability (93.3% accuracy, Fig. 3D). The gap between

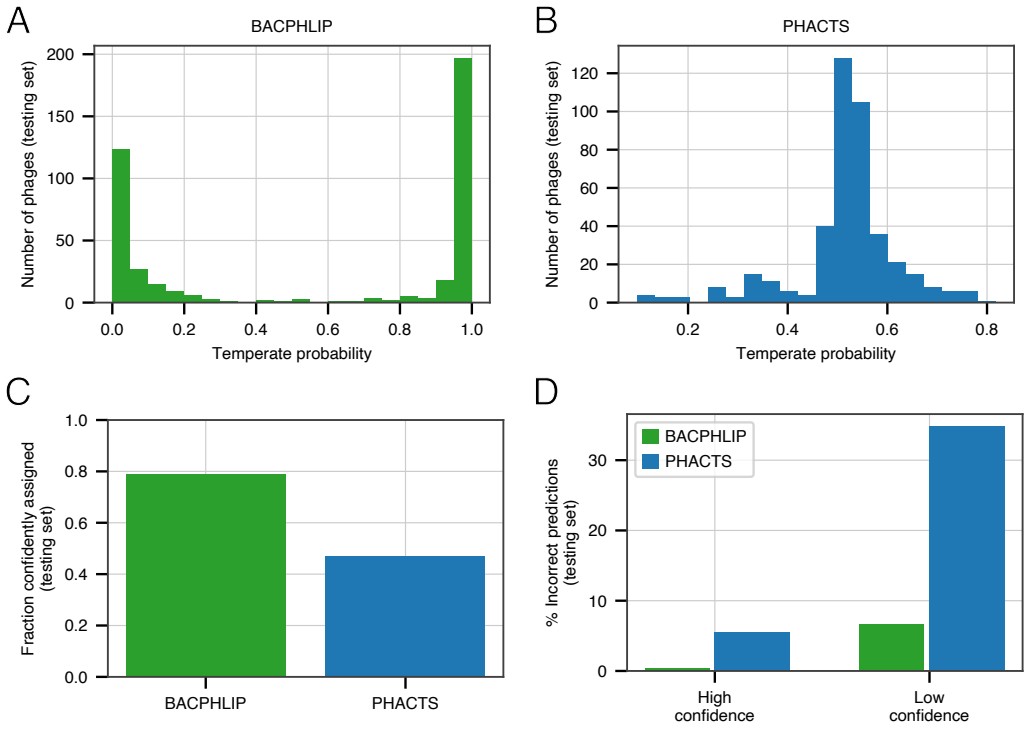

**Figure 3 Evaluating the effect of prediction confidence.** (A) The distribution BACPHLIP predictions (testing set) where a value of 1 corresponds to a high confidence temperate prediction and a value of 0 corresponds to a high confidence virulent prediction. (B) As in (A), showing results for the PHACTS model. (C) The fraction of the testing set ($n = 423$) that is confidently assigned to a lifestyle category by BACPHLIP and PHACTS. (D) Error rates for each method on high and low confidence testing set predictions.

accuracy on high and low confidence predictions was particularly stark for PHACTS: high confidence predictions had an accuracy of 94.4% whereas the accuracy on low confidence predictions was only 65.2% (Fig. 3D).

In order to enrich accuracies and depending on the application, our evaluation strongly suggests that users follow PHACTS guidelines and only rely on high confidence predictions. Users of BACPHLIP may also wish to restrict their analyses to genomes with ≥95% class probability and treat those genomes with lifestyle prediction between 50 and 95% as uncertain.

## DISCUSSION

Across all tested data sets, BACPHLIP substantially outperforms previous methods for classifying phage lifestyles. We leverage several ideas from previous studies, specifically: (i) identifying the presence of a well-defined set of protein domains (*Mavrich & Hatfull, 2017*) and (ii) using machine learning via a Random Forest classifier (*McNair, Bailey & Edwards, 2012*). We unite these ideas in one framework and show that doing so leads to large improvements in the ability to predict whether a given phage is temperate or virulent. Critically, BACPHLIP rapidly creates predictions from un-annotated genome sequences

alone, and thus can facilitate large-scale studies into phage genome evolution (*Moura de Sousa et al., 2021*). Given the rapid expansion of phage genome data sets (*Roux et al., 2021*; *Camarillo-Guerrero et al., 2021*), comparative genomic analyses have the potential to drastically increase our understanding of phage diversity and function.

Our supervised learning-based approach relies on access to training data (phages with known lifestyles) and we encourage all users of BACPHLIP to recognize the contribution made by *Mavrich & Hatfull (2017)* who assembled this dataset. However, we also wish to strongly emphasize several caveats to the existing dataset and supervised-learning models more generally. Most prominently, the existing dataset is made up almost exclusively of phages from within the *Caudovirales* viral order and is further biased towards a small number of hosts (95% infect species within the orders *Actinobacteria*, *Gammaproteobacteria*, and *Bacilli*). We urge caution when performing so-called out-of-sample prediction (i.e. predicting the lifestyle of phages that infect species from outside of these orders). While BACPHLIP can be run on any phage genome sequence (tens of thousands of which are available in current databases) and will provide the user with lifestyle probability values, a lack of experimentally determined phage lifestyles for many species make it difficult to assess the expected accuracy of this (or any) model on a highly diverse and novel set of phage species. Nevertheless, we showed that the accuracy of BACPHLIP exceeds 95% for the most strictly controlled testing set of species that we could identify.

We also further reiterate that BACPHLIP was developed for use on complete phage genomes and performance on fragmented or partially assembled genomes is likely to be substantially degraded; users are strongly encouraged to ensure that the starting assumptions are met prior to running BACPHLIP. This limitation arises because the underlying core of BACPHLIP relies on finding protein domains within the input genome. If a genome is only 50% complete, the lack of lysogeny-associated proteins may be due to the fact that the phage genome is virulent or it may simply be because the relevant domains are encoded within the missing genome segments. Of course, the more complete a genome, the less likely that these errors will arise so users may themselves assign varying degrees of prediction confidence according to genome completeness (which can be assessed with other programs such as CheckV (*Nayfach et al., 2020*)). Additionally, if a genome fragment is predicted to be temperate it is likely that this determination is accurate; it is only for phages categorized as virulent that the issue of potentially missing lysogeny-associated protein domains arises.

Finally, we note that after completion of this work, another phage lifestyle classification tool came to our attention (*Tynecki et al., 2020*). The PhageAI program uses machine learning and natural language processing to classify phage genome sequences, and is released as a web-based platform with an API that allows for programmatic access. Despite using a conceptually distinct and fully-independent approach, the authors of PhageAI have reported nearly identical accuracy values to BACPHLIP—highlighting the robustness of both approaches. It is our expectation that BACPHLIP and PhageAI will perform similarly with regard to lifestyle prediction and serve different niches, with the PhageAI program being largely web-hosted with plans for a larger feature set beyond lifestyle prediction and BACPHLIP providing users with a light-weight option that can be run rapidly and locally.

## CONCLUSIONS

Here, we have shown that a machine learning classifier (BACPHLIP) that relies on the presence/absence of a targeted set of protein domains is capable of predicting phage lifestyle with a level of accuracy that greatly supersedes previously existing software. We anticipate that the accuracy of BACPHLIP will increase in future releases as: (i) more phylogenetically diverse phages become available for training the classifier (potentially via analysis of prophages and/or using information contained in meta-genomic studies) and (ii) discovery and annotation of conserved protein domains improves to encapsulate new protein domains and more phylogenetic diversity amongst the existing domains that BACPHLIP currently relies on. BACPHLIP is provided entirely open-source, as both a python library for creating custom pipelines and as a a comprehensive command-line interface that accepts genome-sequence input in the 'fasta' format and returns highly-accurate lifestyle prediction within seconds.

## ACKNOWLEDGEMENTS

The authors acknowledge valuable feedback and support from members of the Wilke lab.

### Funding

This work was supported by National Institutes of Health grants F32 GM130113 to Adam J. Hockenberry and R01 GM088344 to Claus O. Wilke. The funders had no role in study design, data collection and analysis, decision to publish, or preparation of the manuscript.

### Grant Disclosures

The following grant information was disclosed by the authors:
National Institutes of Health: F32 GM130113, R01 GM088344.

### Competing Interests

Claus Wilke is an Academic Editor for PeerJ.

### Author Contributions

- Adam J. Hockenberry conceived and designed the experiments, performed the experiments, analyzed the data, prepared figures and/or tables, authored or reviewed drafts of the paper, and approved the final draft.
- Claus O. Wilke conceived and designed the experiments, authored or reviewed drafts of the paper, and approved the final draft.

### Data Availability

The BACPHLIP model is available on PyPI and GitHub: https://github.com/adamhockenberry/bacphlip.

Code and Data used to create BACPHLIP is available at GitHub and Zenodo: https://github.com/adamhockenberry/bacphlip-model-dev. and Hockenberry, Adam J, & Wilke, Claus O. (2020). Dataset for: "BACPHLIP: Predicting bacteriophage lifestyle from conserved protein domains" (Version initial-submission) [Data set]. Zenodo. http://doi.org/10.5281/zenodo.4058664.

## Supplemental Information

Supplemental information for this article can be found online at http://dx.doi.org/10.7717/peerj.11396#supplemental-information.

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
