# Peer review of "BACPHLIP: predicting bacteriophage lifestyle from conserved protein domains"

_PeerJ, doi:10.7717/peerj.11396_

## Round 0.1 · original submission · Minor Revisions

Your manuscript has received three very positive reviews, with only minor revisions suggested before the manuscript will be suitable for publication.

Reviewer 1 ·

Basic reporting

Overall, the manuscript is very clear, the tool design is well explained and the benchmarks provided should enable any reader to rigorously use this tool and interpret its results.

Experimental design

Hockenberry & Wilke present a new tool (BacPhlip) to identify temperate from virulent phages base on their genome sequence. BacPhlip relies on the detection of marker genes tied to a Random Forest Classifier, which is a robust experimental design.

Validity of the findings

BacPhlip seems to work as well as could be hoped/expected. It is also available on github, and easy to download and run. I thus only have a few minor comments about the text itself.

Additional comments

Minor comments:

l. 10: “Bacteriophage species” should be probably be “Bacteriophages”, as infection cycles are often determined for individual phages, not necessarily a whole species.

l. 10: “temperate (lysogenic) and virulent (lytic)”. Since temperate phages are defined by their ability to enter both lysogenic and lytic cycles, I would suggest moving the “lysogenic” and “lytic” parenthesis to the next sentence, i.e.: “temperate and virulent. Temperate phages are capable of a latent phase of infection within a host cell (lysogenic cycle), whereas virulent phages directly replicate and lyse host cells upon infection (lytic cycle).” (the same is true for l 31-32).

l. 29: “meta-genomics” should be “metagenomic”

l. 68: “errors” may be more clear as “erroneous annotation” (if I understood the intended meaning correctly).

l. 79: “that was” should be “that were”

l. 139: for non-specialist, “lysogeny-associated domains” may be more clear as “lysogeny-associated protein domains”

l. 194: “to detect” would be more clear as “to be detected”

·

Basic reporting

The manuscript is clearly written, and describes the problem, previous solutions, and the gap in knowledge that they aim to fill.

Figure 1 is clear and appropriate, although the figure is the inverse of the text describing it: the paragrap discuss classification accuracy (e.g. BACPHLIP is 98.3% accurate) while the figure shows inaccuracy (% incorrect predictions).

Figure 2 is less informative, and it does not relate to the information in the text describing the figure. For example, the text (lines 202-209) state the "Top 50 protein domains ... accounted for 85% of the model weights" but it is impossible to see that from Fig. 2.

The manuscript demonstrates the utility of BACPHLIP in predicting the lifestyle of temperate and virulent phages, an area of increasing concern.

Experimental design

The manuscript carefully defines the problem and identifies the data set and approach that will be used for the solution. The manuscript combines data from Mavrich and Hatfull with the Random Forests from McNair et al to idenitfy the lifestyle of phages.

The main difference between this approach and McNair is the use of a selected set of domains that are likely to be invovled in lysogeny. McNair used all the proteins in the genome, and thus discovered proteins that may be associated with lysogeny. However, as noted in the manuscript, the approach applied here has identified phages whose mechanism of lysogeny is apparently unknown and novel.

Validity of the findings

The authors took every effort to crouch their results in the appropriate caveats that may affect their findings. They are careful both in the manuscript and in their git repository to remind users that this tool has not beeen designed to work with fragmented/incomplete genomes, and to acknowlege the inherent bias in the data set that they used to train their models.

Given those caveats, BACPHLIP shows remarkable progress in classifying the lifestyle of phages over previous work.

In my experience, I was able to run BACPHLIP and generate meaningful results.

Additional comments

This is the next step advancement over the previous work, that integrates many more genomes to result in higher accuracy.

·

Basic reporting

The manuscript by Hockenberry and Wilke describes the tool BACPHLIP. Overall, the manuscript is very well written and the results are also convincingly and clearly presented in a logical way, with the exception of one of the figures where the data does not seem to match the text (see "general comments"). I have also downloaded and used BACPHLIP. Its usage is fairly intuitive and straightforward, and I had no issues obtaining a lifestyle classification for a dataset of phage genomes.

The literature is well referenced, but one of the points in the introduction would benefit from further detail. This referes to lines 32-33 ("Phages are extraordinarily diverse and this dichotomy is an over-simplification"). I concur, but perhaps this could be expanded a bit more. Are the authors referring to virulent phages that can remain in pseudo-lysogeny within a cell? To phage-plasmids that blur the lines between mobile elements?

There is also an issue (possibly a typo) in Figure 1. The number of phages in the independent dataset is shown as 157, but shouldn't it be 172 phages (line 129 and line 153)? Or did I miss something? If it is a typo, check also the number of related phages (266), since 172+266 does not match the 423 phages used to test the model(s).

Finally, the raw data (protein domains used for classification) is included, but it is not referenced in the main text (at least nowhere I could find).

Experimental design

BACPHLIP performs the very difficult, but necessary task of assigning a lifestyle to phages based on their genomic features. The authors illustrate well the need for this type of classification, specifically based on the fact that the only other available tool was based on a limited set of genomes. The methodology developed to achieve this goal is clear and, as far as I can discern, looks robust.

The authors provide the protein domains that underlie the Random Forest model, and also provide the code to test and build the classifier in BACPHLIP (although I have not tested this specific code), which is great for users that want to re-train the model.

Validity of the findings

The authors find, and convincingly show, that BACPHLIP outperforms other approaches to classify phage lifestyle. They also control for a potential statistical inflation of their results by taking into account phage relatedness in an additional analysis.

Additional comments

Overall, I enjoyed the manuscript and I find it very close to be suitable for publication. My main point is not even really a criticism, but it concerns the link between BACPHLIP and the dataset of Mavrich and Hatfull. The authors mention that (lines 176-177) "[the] similarity [between BACPHLIP and Mavrich] is perhaps not surprising given that both of these methods rely on targeted identification of protein domains associated with lysogeny". But in fact the two methods are linked in a more intrinsically fashion, because the model beyond BACPHLIP was trained by using the lifestyle assignment of Mavrich, and not just because they use the same type of genomic features. That is, if I understood correctly, any misclassification of (e.g.) a temperate phage in Mavrich will have an impact in how the RF model is scored during training, which will itself influence future lifestyle assignments in the testing dataset.

To be clear, I don't see this as a major issue, or at least not one that can be resolved. It is already explicit in the methods that the Mavrich dataset underlies the training of the model, and there really is a lack of other sources of data for phage lifestyle (it is impossible to experimentally or manually classify them all). However, I do feel it is important to reinforce this (perhaps in the discussion?) as an idiosyncrasy of the method, beyond the biased phylogenetic composition of the Mavrich dataset (lines 244-246).

There are two other suggestions that I think would improve the manuscript, but neither is critical:

1) It could be interesting to show how PHACTS fares when considering only phages whose lifestyle was classified as "confidently" assigned. Is the inaccuracy vastly different? Note that I am aware that PHACTS very often results in non-confident assignments, which by itself is an issue. But this could be important to understand a) whether previous works that used this classification but limited their results to the "confidently assigned" phages are robust, and b) to further reinforce that BACPHLIP outperforms even the most stringent PHACTS classification (if that is the case). I think a supplementary figure/table, or even a mention of this additional analysis in a sentence in the text could be informative.

2) Regarding the prediction confidence, the authors state that 333 genomes had >=95% probability for a given lifestyle, with only one error in this set, which is quite good. However, I wonder how this distribution looks like for the whole testing dataset. Is it binomial, i.e., do the probabilities cluster around 95% for those 333 genomes and then at low values (~50%) for the remainder genomes? Or is it more evenly distributed? And where do the lifestyle mis-assignments tend to occur in this distribution? The authors suggest that the users of BACPHLIP should treat predictions between 50% and 95% as uncertain, but being aware of this distribution could help them make a more informed decision regarding the cutoffs for significance.

Additionally, I have a couple of two other minor points:

1) Line 62: "parA|ParA|parB|ParB". I was initially a bit confused as to why these term were used as genomic feature. I had to search in the literature to find out that they refer to the partitioning systems of actinobacteriophage, which was unknown to me. A sentence on what are these genes, and why are they potentially relevant to discriminate temperate from virulent phages would be useful for other readers.

2) Lines 221-223: "However, we note that varying degrees of confidence are (and should be placed by users) on the predicted lifestyle of individual phages that have 51% vs 99% probability, for instance." The phrasing of the sentence seems confusing to me. Rephrase?

---

## Round 0.2 · accepted · Accept

Thank you for carefully addressing the reviewers' minor feedback.